# Epidemiological Features of Human Norovirus Genotypes before and after COVID-19 Countermeasures in Osaka, Japan

**DOI:** 10.3390/v16040654

**Published:** 2024-04-22

**Authors:** Tatsuya Shirai, Juthamas Phadungsombat, Yumi Ushikai, Kunihito Yoshikaie, Tatsuo Shioda, Naomi Sakon

**Affiliations:** 1Department of Microbiology, Osaka Institute of Public Health, Osaka 537-0025, Japan; shirai@iph.osaka.jp (T.S.);; 2Research Institute for Microbial Diseases, Osaka University, Osaka 565-0871, Japan; juthamas@biken.osaka-u.ac.jp

**Keywords:** norovirus, dual typing, molecular epidemiology, COVID-19

## Abstract

We investigated the molecular epidemiology of human norovirus (HuNoV) in all age groups using samples from April 2019 to March 2023, before and after the COVID-19 countermeasures were implemented. GII.2[P16] and GII.4[P31], the prevalent strains in Japan before COVID-19 countermeasures, remained prevalent during the COVID-19 pandemic, except from April to November 2020; in 2021, the prevalence of GII.2[P16] increased among children. Furthermore, there was an increase in the prevalence of GII.4[P16] after December 2022. Phylogenetic analysis of GII.P31 RdRp showed that some strains detected in 2022 belonged to a different cluster of other strains obtained during the present study period, suggesting that HuNoV strains will evolve differently even if they have the same type of RdRp. An analysis of the amino acid sequence of VP1 showed that some antigenic sites of GII.4[P16] were different from those of GII.4[P31]. The present study showed high infectivity of HuNoV despite the COVID-19 countermeasures and revealed changes in the prevalent genotypes and mutations of each genotype. In the future, we will investigate whether GII.4[P16] becomes more prevalent, providing new insights by comparing the new data with those analyzed in the present study.

## 1. Introduction

Human norovirus (HuNoV) is a causative agent of acute gastroenteritis worldwide [1]. HuNoV is spread by contact with contaminated food or infected individuals and can infect people of all ages, causing an estimated 200,000 deaths and a global economic burden of approximately $60 billion annually [2,3]. Norovirus belongs to the Caliciviridae family and has a positive-sense and single-stranded RNA genome of 7.5–7.7 kb but lacks an envelope. The genome comprises three open reading frames (ORFs), except for murine noroviruses, which contain a fourth ORF. ORF1 encodes six nonstructural proteins, including RNA-dependent RNA polymerase (RdRp), and ORF2 and ORF3 encode the capsid proteins, namely viral proteins 1 (VP1) and 2 (VP2), respectively.

The noroviruses detected in animals are classified into 10 genogroups (GI–GX) and >48 genotypes based on the amino acid sequence of the capsid (genotyping) and are also classified into eight P-groups and >60 P-types based on the nucleotide sequence of RdRp (P-typing) [4]. Norovirus GI and GII are frequently detected genogroups in humans worldwide. The recombination between ORF1 and ORF2 is reported to be a pandemic-associated factor [5,6], and genotype and P-type analyses (dual typing) are recommended [4]. The functional motifs of RdRp have been reported for motif A (residue 240–251), B (residue 297–311), C (residue 336–348), D (residue 366–370), E (residue 389–394), F (residue 160–172 and 181–187), and G (residue 112–121) [7].

HuNoV GII.4 is the most prevalent genotype currently known, and the emergence of new variants of GII.4 has caused pandemics and the replacement of variants [8,9]. The most recent variant reported to have caused a pandemic was that of Sydney 2012 [10]. For HuNoV GII.4, the antigenic sites A–H of VP1 have been determined [8,11,12], and the mutation patterns determined for three antigenic sites (A, C, and G) correlated well with the circulation of GII.4 variants. However, neither antigenic site B nor H correlated with the circulation of the GII.4 variant [9].

Non-GII.4 pandemics were reported in the 2010s, one of which was caused by GII.2[P16]. GII.2[P16] was reported in Japan in 2009 [13] and has also caused outbreaks in Japan, China, Canada, and Germany since mid-2016 [14,15,16,17]. The VP1 region of the 2016 pandemic strain had no characteristic amino acid substitutions compared to those found in GII.2 detected before 2016, suggesting that immune escape and altered affinity for tissue blood group antigens were not involved in the pandemic [18,19]. In the RdRp region, five amino acid substitutions (D173E, S293T, V332I, K357Q, and T360A) are characteristic of GII.2[P16] and GII.4[P16] detected after 2016 [18,20], but their impact on the pandemic in 2016 remains to be investigated.

The GII.4 Sydney variant contained a GII.P31 RdRp when first detected in 2012 [21,22]. However, a recombinant strain, GII.4 Sydney with GII.P16 RdRp, emerged in 2016 [16,17,23]; this strain replaced GII.2[P16] globally [24,25,26,27,28]. Consequently, GII.4[P16] was the most frequently detected global strain before the start of COVID-19 countermeasures [29] but was less frequently detected in East Asia, including Japan, than in other regions [30,31].

The COVID-19 countermeasures affected the occurrence of many other viral infectious diseases in Japan. In April 2020, the first state of emergency was implemented in Japan. The effects of this intervention were evident for infectious gastroenteritis and foodborne viral outbreaks, and the number of reports of these diseases in 2020 was the lowest since 2011 [32].

Since 2021, the number of HuNoV detections worldwide has been gradually increasing because of the reduction in COVID-19 countermeasures. In the context of such a change, the human-to-human pediatric outbreaks from 2020 to 2021 in Shanghai were reported to be caused by the GII.4[P16] sublineage [30,33]. We investigated the impact of COVID-19 countermeasures on HuNoV genotypes and genetic mutations in Japan, which had a different detection status from other countries even before COVID-19, including a low detection of GII.4[P16].

## 2. Materials and Methods

### 2.1. Sample Collection

Human stool samples were collected from April 2019 to March 2023 in Osaka Prefecture, Japan, under the three surveillance systems previously described [34] and were allocated to four groups: (A) sporadic pediatric acute gastroenteritis cases (<15 years of age), (B) human-to-human acute gastroenteritis outbreaks in childcare and educational facilities, (C) suspected foodborne outbreaks, and (D) human-to-human acute gastroenteritis outbreaks in nursing homes. For sporadic cases, 326 samples were analyzed. For human-to-human outbreaks in childcare and educational facilities, 801 samples from 258 outbreaks were analyzed. In addition to children, we analyzed HuNoV from adults, such as childcare workers, but only for outbreaks in childcare facilities or elementary schools attended by children <15 years of age. For suspected foodborne outbreaks, 690 samples from 139 outbreaks were analyzed; outbreaks in childcare facilities or elementary schools attended by children aged < 15 years were omitted. Since samples were routinely tested the day after they had been received, the day before the test date was considered the collection date if the collection date was unknown. For human-to-human outbreaks in older adults in nursing homes, 105 samples from 34 outbreaks were analyzed.

### 2.2. Detection of HuNoV

For each sample, a 10% stool suspension was prepared with phosphate-buffered saline, then thoroughly mixed via vortexing, and clarified via centrifugation at 15,000 rpm for 5 min at 4 °C. Nucleic acids were extracted from 200 μL of the supernatant using the MagDEA Dx SV with magLEAD 12gC equipment (Precision System Science, Chiba, Japan). Real-time quantitative PCR was performed in 20 μL of a reaction mixture containing 2 μL of RNA, 5 μL of TaqMan Fast Virus 1-Step Master Mix (Applied Biosystems, Foster City, CA, USA), a 400 nM concentration of each primer (GI: COG1F and COG1R, GII: COG2F and COG2R), and either 5 pmol of RING1(a)-TaqMan Probe (TP) (FAM-ATYGCGATCYCCTGTCC-MGB) and 5 pmol of RING1(b)-TP (FAM-ATCGCGGTCTCCTGTCC-MGB) for HuNoV GI detection or 5 pmol of RING2AL-TP (FAM-TGGGAGGGSGATCGCRATCT-MGB) for HuNoV GII detection [35,36,37]. PCR amplification was performed using a StepOne Plus real-time PCR system or a Quantstudio 5 real-time PCR system as follows: incubation at 50 °C for 5 min and initial denaturation at 95 °C for 20 s, followed by 40 cycles of amplification with denaturation at 95 °C for 15 s and annealing and extension at 56 °C for 1 min.

### 2.3. Sequencing and Genotyping

At least one positive sample, representative of each HuNoV outbreak, was selected for the sequence analysis of the partial RdRp and VP1 genetic regions. HuNoV RNA was amplified via RT-PCR with the Mon432/G1SKR and Mon431/G2SKR primers (579 and 570 bp, respectively) [38]. RT and PCR reactions were performed using MuLV Reverse Transcriptase (Life Technologies, Carlsbad, CA, USA) and KOD multi & Epi (Toyobo, Osaka, Japan), respectively. A positive sample was selected from 21 GII.P16- and 10 GII.P31-positive outbreaks and analyzed for the whole RdRp-coding region. One or two positive samples were selected from 17 GII.2- and 13 GII.4-positive outbreaks and analyzed for the whole VP1-coding regions. The RdRp region of GII.2[P16] was amplified via RT-PCR with the GII.2P16_3530F (5′-ATCTGTGCCACACAGGGAAG-3′) and GII.2P16_5132R (5′-GGCTGCACCATCAGTAGATG-3′) primers (1.6 kbp). The RdRp region of GII.4[P16] was amplified with the GII.2P16_3530F and GII.4P31_5132R (5′-GGCTGCAGACCCATCAGATG-3′) primers (1.6 kbp). The RdRp region of GII.4[P31] was amplified with the GII.4P31_3463F (5′-CAAGAGGGGGAATGACTACG-3′) and GII.4P31_5132R primers (1.6 kbp). The VP1 region was amplified with COG2F and TX30SXN primers (2.6 kbp) [39]. RT-PCR for the whole RdRp or VP1 regions was performed using PrimeScript One Step RT-PCR Kit Ver.2 (Takara Bio Inc., Shiga, Japan). PCR products were sequenced using the BigDye Terminator Cycle Sequencing Kit v1.1 or v3.1 (Applied Biosystems).

The HuNoV genotype/P-type combination was identified with the Norovirus Genotyping Tool version 2.0 (https://www.rivm.nl/mpf/typingtool/norovirus/ [accessed on 7 February 2024]), supplied by NoroNet, or with the Human Calicivirus Typing Tool (https://calicivirustypingtool.cdc.gov/ [accessed on 7 February 2024]), supplied by the Centers for Disease Control and Prevention, USA. The sequences obtained in the present study were deposited in the GenBank database under the following accession numbers: LC798000–LC798042 and LC798257–LC798258. Four GII.4 sequences, namely LC645995–LC645998, which were collected during the present study period and had been reported in the previous study [40], were also included in the present analysis.

### 2.4. Preparation of HuNoV Nucleotide Sequence Dataset

Reference sequences representing either genotypes or P-types were collected from the Human Calicivirus typing tool. Other sequences similar to our HuNoV sequences were collected by using the Basic Local Alignment Search Tool (https://blast.ncbi.nlm.nih.gov/blast/Blast.cgi/ [accessed on 13 February 2024]). All sequences analyzed in the present study were retrieved from GenBank, the NCBI nucleotide database (accessed on 13 February 2024). The HuNoV genotypes and P-types were confirmed by the Human Calicivirus typing tool before proceeding with further analyses.

Four nucleotide datasets were prepared for molecular clock analysis, including (1) the Japan GII.P16 RdRp dataset, consisting of 100 RdRp sequences (21 from the present study, 7 from the outside of the areas covered by our surveillance systems, and 72 from GenBank) and representing the GII.P16 norovirus associated with GII.17, GII.2, and GII.4 types, which were detected in Japan from 2001 to 2023; (2) the Japan GII.P31 RdRp dataset, comprising 84 RdRp sequences (11 from the present study and 73 from GenBank, collected from 2007 to 2022), of which, 78 sequences were associated with GII.4 Sydney and 6 were associated with GII.4 Osaka variants; (3) the GII.2 capsid dataset, comprising 80 GII.2 Japan sequences (13 from the present study, 4 from the outside of the areas covered by our surveillance systems, and 63 from GenBank), specifically those of GII.2[P16], which circulated between 2008 and 2022; and (4) the GII.4 capsid dataset, including 104 VP1 sequences (11 from the present study, 4 from the previous study [40], 3 from the period before the present study, and 86 from GenBank), representing 10 epidemic variants from various regions of Japan, which circulated from 1998 to 2022.

### 2.5. Phylogenetic Tree Analyses

Nucleotide datasets were aligned using AliView v1.28 [41]. Each dataset was screened for recombination breakpoints using the Genetic Algorithm for Recombination Detection performed at https://www.datamonkey.org/gard (accessed on 15 February 2024) [42]. A maximum-likelihood phylogenetic tree and the substitution model selection were performed using IQ-TREE with 1000 ultrafast bootstraps [43]. The time-scaled tree with the time of the most recent common ancestor (tMRCA) and the rate of evolution (substitution/site/year, s/s/y) were estimated based on the Bayesian Markov chain Monte Carlo (MCMC) approach employed in BEAST packages v1.10.4 [44]. Before the Bayesian analysis, datasets were determined for an evolutionary temporal signal through root-to-tip analysis using Tempest v1.5.3 [45]. An uncorrelated lognormal relaxed molecular clock and Bayesian Skygrid were used for evolutionary estimation [46]. Each dataset was run in duplicate with MCMC with 50 million chains and sampling every 5000 generations. The trace of individual runs, examined in Tracer v1.7.1, demonstrated effective sample sizes exceeding 200 [47]. Runs were combined using LogCombiner v1.10.4 with burn-in at 10%. The resulting maximum clade credibility (MCC) tree was generated using TreeAnnotator v1.10.4 and visualized in Figtree v1.4.4 (http://tree.bio.ed.ac.uk/software/figtree). Skygrid demographic was reconstructed in Tracer v1.7.1.

### 2.6. Stastiscal Analyses

The numbers of HuNoV-positive cases (0–11 y) detected from April to September 2020 were compared to those in the previous year, from April to September 2021, using a one-tailed *t*-test at a significance level of *p* < 0.05.

### 2.7. Ethical Approval

The study protocol was approved by the ethics committee of the Osaka Institute of Public Health (No. 0710-03-6).

## 3. Results

### 3.1. HuNoV Dual Typing in Each Group

Samples were categorized into four groups depending on the surveillance approach and mode of infection (Table 1). Group A included 112 HuNoV-positive samples from sporadic pediatric gastroenteritis cases (<15 years of age). Group B included 441 HuNoV-positive samples from 182 outbreaks of infectious gastroenteritis in childcare facilities and schools. Group C included 185 HuNoV-positive samples from 50 suspected foodborne outbreaks. Group D included 89 HuNoV-positive samples from 32 infectious gastroenteritis outbreaks in older adults in nursing homes (Table 1).

Genotype/P-type combinations were counted for sporadic pediatric cases (Group A) in each case, human-to-human outbreaks (Groups B and D), and suspected foodborne outbreaks (Group C) in each outbreak (Figure 1). Genotypes were determined for 112 samples in Group A, 387 samples in Group B, 109 samples in Group C, and 64 samples in Group D. If two genotype/P-type combinations were detected in an outbreak case, both were counted. Duplicate genotype/P-type combinations were not detected in sporadic cases. GII.4[P31] was the most frequently detected type in Groups A and D, and GII.2[P16] was the most frequently detected type in Groups B and C. GII.2[P16] and GII.4[P31] were the top two types in Groups A to C. GII.4[P16] and GII.3[P12] were frequently detected following the top two types in Groups A and B, whereas GII.17[P17] and GII.4[P16] were frequently detected in Group C.

### 3.2. Genotype/P-Type Combinations of HuNoV Detected in Osaka before and after COVID-19 Countermeasures

Monthly counts of the number of HuNoV-associated acute gastroenteritis cases or outbreaks reported in Osaka from April 2019 to March 2023 were prepared, focusing on six major genotype/P-type combinations: GII.2[P16], GII.3[P12], GII.4[P16], GII.4[P31], GII.6[P7], and GII.17[P17] (Figure 2). A total of four states of emergency were declared in Osaka because of the COVID-19 pandemic. Following the first state of emergency, the number of HuNoV detections was within two cases or outbreaks in all groups from April to November 2020. In March 2021, after the end of the second state of emergency, the number of GII.2[P16] cases or outbreaks detected in children (Groups A and B) was highest during the present study period. In addition, the number of suspected foodborne outbreaks (Group C) caused by GII.2[P16] was highest in April 2021. Before November 2022, GII.4[P16] was not detected in multiple cases in a month in Groups A–C, but GII.4 [P16] was detected in multiple cases or outbreaks in Group A from December 2022 to February 2023, in Group B from January to February 2023, and in Group C in January and March 2023. Conversely, in human-to-human outbreaks in older adults (Group D), only one GII.2[P16] outbreak was detected since January 2021, and GII.4[P16] had not been detected since March 2020.

### 3.3. Distribution of Genotype/P-Type Combinations in Children at Each Age

The distribution of genotypes and P-types of HuNoV in children aged from 0 to 11 years was investigated using datasets of each sporadic case and each child in human-to-human outbreaks (Figure 3). From April to September 2019, multiple types were detected in 0–2-year-old infants, but GII.2[P16] was most frequently detected in children older than 4 years old. From October 2019 to March 2020, GII.4[P31] was detected in all age groups and was predominant in 1–3-year-old children. From April to September 2020, the number of HuNoV detections decreased compared to the same period in the previous year because of the COVID-19 countermeasures, with only four cases reported (*p* < 0.05). From October 2020 to March 2021, GII.2[P16] was detected in all age groups and predominated until September 2021. However, from October 2021 to March 2022, almost all HuNoV strains detected were GII.4[P31], and GII.2[P16] was not detected. Consequently, the main epidemic strain clearly changed from GII.2[P16] to GII.4[P31]. No HuNoV type was clearly dominant from April to September 2022, but GII.3[P12] was detected in all age groups. GII.2[P16], GII.4[P16], and GII.4[P31] were all detected from October 2022 to March 2023 at each age from 0 to 5 years.

### 3.4. Phylogenetic Analysis of RdRp and VP1 of HuNoV

A total of 36 samples from our surveillance were analyzed for their nucleotide sequences of the whole RdRp- and VP1-coding regions. We were able to determine 33 RdRp- and 28 VP1-coding regions. These included 21 GII.P16, 11 GII.P31, 13 GII.2, and 15 GII.4. An additional seven samples collected during the present study period in Osaka but outside the areas covered by the surveillance systems were also analyzed for their nucleotide sequences and were included as reference sequences. These included seven GII.P16 and four GII.2. An additional three GII.4 samples collected before the present study period in Osaka were also analyzed for VP1 nucleotide sequences and were included as reference sequences.

Phylogenetic analyses of RdRp with root-to-tip analysis were performed, and the linear regression of divergence versus collection time showed a positive temporal signal of 0.89 and 0.90 for the GII.P16 RdRp and GII.P31 RdRp dataset, respectively (Figure 4A and Figure 5A). Starting from 2008, the effective population size (Ne) of GII.P16 RdRp increased slightly on four occasions: mid-August 2010 (2010.61), late March 2014 (2014.23), briefly in late December 2019 (2019.90), and in late December 2021 (2021.97) after a sudden drop (Figure 4B). The MCC tree of GII.P16 shows that GII.2[P16] collected after 2016 formed a distinct cluster, clearly separated from those collected before 2015 (Figure 4C). The GII.2[P16] cluster of viruses after 2016 had a tMRCA at 2012.52 (95% highest posterior density (HPD) interval: 2011.34–2013.69) with a posterior probability (PP) of 0.89 and an evolutionary rate of 2.21 × 10^−3^ (95% HPD interval: 5.87 × 10^−4^ to 4.15 × 10^−3^) s/s/y (Figure 4C). However, the GII.4[P16] cluster emerged at 2013.37 (95% HPD interval: 2012.08–2014.68) and had an evolutionary rate of 2.82 × 10^−3^ (95% HPD interval: 2.14 × 10^−3^ to 5.50 × 10^−3^) s/s/y (Figure 4C). From 2010 to the current time, the Ne of GII.P31 RdRp showed two phases of increase, starting in early February 2010 (2010.10) and mid-April 2016 (2016.29), respectively. The GII.P31 Ne was steady from late May 2019 to the end of December 2019 and then decreased until 2023, the time of this analysis (Figure 5B). GII.P31 associated with the GII.4 Sydney variant had an estimated tMRCA at 2007.15 (95% HPD interval: 2004.82–2009.89) and an evolutionary rate of 3.55 × 10^−3^ (95% HPD interval: 1.02 × 10^−4^ to 1.13 × 10^−2^) s/s/y (Figure 5C). Nine of the newly obtained GII.P31 RdRp strains from Osaka clustered to strains from Tokyo and Shizuoka collected between 2016 and 2023, sharing the tMRCA of 2014.95 (95% HPD interval: 2014.08–2015.90). Another two strains, namely 22-012 (LC798034) and 22-013 (LC798013), were included in a different cluster and were related to a Kyoto strain collected from 2014 to 2015.

Root-to-tip analysis in the VP1 phylogenetic analyses showed a positive clock-like correlation coefficient of 0.88 and 0.94 for the GII.2 and GII.4 datasets, respectively (Figure 6A and Figure 7A). GII.2 viruses associated with GII.P16 had been circulating in Japan from 2008 to the time of analysis. The GII.2 VP1 Ne frequently fluctuated from 2008 to 2023 (Figure 6B), peaked in 2013, and spiked several times in mid-April 2016, early May 2018, and mid-December 2020 (2016.29, 2018.35, and 2020.94, respectively). The MCC tree (Figure 6C) revealed that GII.2 strains detected after 2016 formed a distinct clade (PP = 1), with an estimated tMRCA of 2011.46 (95% HPD interval: 2009.36–2014.01). The evolutionary rate was estimated at 1.98 × 10^−3^ (95% HPD interval: 7.24 × 10^−4^ to 3.46 × 10^−3^) s/s/y. Within this clade, the Osaka strains clustered with the Tokyo strains collected during the same period (2018–2023). However, two distinct clades with a tMRCA of 2015.72 and 2016.72 could be observed. For GII.4, 10 variants with various P-types have been circulating since 1998 (Figure 7C). GII.4 Sydney 2012 is the current circulating variant, which was introduced to Japan around 2007.76, and was subsequently separated into two distinct clades based on different P-types. The GII.4 Sydney with GII.P31 clade comprised viruses collected from 2011 to 2023, sharing a tMRCA of 2008.72 (95% HPD interval: 2007.62–2009.90) and an evolutionary rate of 2.90 × 10^−3^ (95% HPD interval: 9.77 × 10^−4^ to 5.19 × 10^−3^) s/s/y. The newly obtained Osaka strains from 2019 to 2023, along with other strains from Tokyo, Kyoto, and Shizuoka, collected in 2016 and from 2019 to 2023, formed a subcluster distinct from the 2011–2016 strains and had a tMRCA of 2012.92 (95% HPD interval: 2011.42–2014.30). The GII.4[P16] clade, comprising Kawasaki, Osaka, Aichi, and Tokyo viruses collected from 2016 to 2023, had a tMRCA at 2013.54 (95% HPD interval: 2012.54–2014.62) and an evolutionary rate of 3.68 × 10^−3^, 95% HPD interval: 2.14 × 10^−3^ to 5.50 × 10^−3^) s/s/y. The newly obtained Osaka strains from 2020 and 2023 clustered with recent Tokyo strains from 2023, sharing a tMRCA of 2015.75 (95% HPD interval: 2014.55–2016.90). The Ne trend of the GII.4 Sydney variant remained steady from mid-February 2011 to mid-April 2015 (2011.13–2015.26) and then slightly increased from October 2015 to mid-May 2018 (2015.77–2018.35).

### 3.5. Analysis of RdRp Amino Acid Sequences and VP1 Antigenic Sites

A total of 510 amino acid sequences of GII.P16 RdRp were compared among strains obtained from April 2019 to March 2022 (Table 2). For GII.2[P16], 14 amino acid mutations existed among strains obtained in the present study, but no sites showed a unique change before or after the COVID-19 countermeasures. There were 11 amino acid mutations among the GII.4[P16] strains obtained in the present study. The two GII.4[P16] strains in 2020 (19S55 and 19S57) had only one substitution at residue 427 compared to the reference strain from Shanghai in 2021, and the frequency of substitution against the reference strain from Shanghai in 2021 was the lowest among the GII.4[P16] strains in the present study. At residue 457, all the GII.2[P16] strains contained Lys (K), while all the GII.4[P16] contained Arg (R). Motif G mutations were detected in two strains of GII.2[P16] at residue 121. No other amino acid mutations were detected in the motifs.

The amino acid sequences were analyzed for the GII.4 VP1 antigenic sites (Table 3). For the GII.4[P31] antigenic site, seven out of eight strains contained the peptide sequence TGSHNENN in antigenic site A, and only one strain detected in 2022 contained TGSHNENH. No difference was found for antigenic sites C, D, and G among GII.4[P31] strains obtained in Osaka. For the GII.4[P16] antigenic site, five out of seven strains contained TGSHNENH in antigenic site A, and two strains detected from the same outbreak contained TGSRNEDH. These two GII.4[P16] strains also contained a unique antigenic site C of RTDFEVN, while this was RTDFEAN in five other strains. Antigenic sites D, E, and G did not differ among GII.4[P16] strains collected in Osaka. Comparing GII.4[P31] to GII.4[P16], TGSHNENN was predominant in antigenic site A in GII.4[P31], whereas TGSHNENH was predominant in GII.4[P16]. Antigenic sites D and E differed between GII.4[P31] and GII.4[P16], but only one strain of GII.4[P31] in 2022 contained a sequence of SRSTP at antigenic site E.

## 4. Discussion

The samples collected in the present study were allocated to Groups A–D. Group D, which mainly comprised older adults, showed a significantly different trend compared to Groups A–C, such as no increase in detection of GII.2[P16] in 2021 and no detection of GII.4[P16] in 2023 (Figure 2). The investigation of HuNoV in older adults was limited as most nursing homes have a full-time doctor, and the immunochromatography test for HuNoV is covered by health insurance, as prescribed by the Ministry of Health, Labor, and Welfare. Consequently, few samples were submitted for testing by regional health centers, and insufficient samples may have been available to understand the HuNoV epidemics in older adults. However, the differences between Group D and the other groups were possibly caused by decreased contact with outsiders to prevent COVID-19 outbreaks in older adults, who can manifest severe symptoms of COVID-19.

The GII.2[P16] epidemic in children in 2021 was notable for two points. First, the resurgence of GII.2[P16] in Japan was noteworthy, despite the decline in other countries [33,48]. Second, the number of HuNoV detections in March 2021 was highest during the present study period in Groups A and B. Although restrictions on social activities under the first state of emergency declaration could suppress HuNoV epidemics, thereafter, HuNoV detection increased. This increasing trend was generally consistent with the spike observed in GII.2 Ne in mid-December 2020 (2020.94) (Figure 6B). The switching of the predominance of GII.2[P16] and GII.4[P31] clearly occurred every other year, especially in Groups A and B (Figure 2 and Figure 3). Previous reports from Japan have also suggested a seasonal change in dominant genotypes [34,49]. Genotype-specific immunity to HuNoV could last approximately three years [34,50], which could contribute to the switching of predominant types.

We previously reported that the trend in genotypes in infants between 0 and 2 years old differed from that in children >3 years old [34]. However, for the study period (October 2020 to March 2022), the predominant genotype/P-type combinations were matched among the ages of 0–5 years (Figure 3). This was most likely caused by a change in contact patterns, with decreased contact at nursery school and increased contact at home, because of the COVID-19 countermeasures reducing the number of days or shortening the length of time at nursery school. The genotype/P-type combinations in adults during the same period were also similar to those in children (Figure 2).

In all groups, from April 2019 to November 2022, the frequency of GII.4[P16] remained lower than that of GII.4[P31] or GII.2[P16] (Figure 2). This trend did not differ from that reported in Japan from July 2018 to June 2021 [31] but was different from that observed elsewhere, such as in Germany [48]. However, since December 2022, a higher number of GII.4[P16] cases/outbreaks have been detected in Groups A–C. This was generally consistent with the results showing that the Ne of GII.P16 and GII.4 increased around the same time: since mid-November 2021 (2021.97; Figure 4B) and since late June 2022 (2022.48; Figure 7B). The phylogenetic analysis showed that the GII.4[P16] detected in Osaka in 2023 (LC798019, LC798020, LC798024, LC798035, LC798037, and LC798038) belonged to a different cluster from that detected in Osaka in 2020 (LC798003, LC798004, and LC798029) and in the Shanghai 2020–2021 cluster (OQ940069 and OQ940080) [30] (Appendix A). Analyses of the amino acid sequences of RdRp and the antigenic sites of VP1 also showed that the Shanghai strain was closer to the GII.4[P16] strains detected in Osaka in 2020 than those in 2023, indicating that an epidemic of GII.4[P16] in 2023 in Osaka occurred via different routes (Table 2 and Table 3). An analysis of the GII.P16 RdRp showed no change in the five amino acids characteristic of GII.2[P16] and GII.4[P16] detected after 2016 [18]. Furthermore, no major changes occurred in the motif sites, suggesting no functional changes (Table 2).

No major antigenic changes in GII.4[P31] or GII.4[P16] occurred before or after the four states of emergency for COVID-19, respectively. However, antigenic sites A, D, and E in VP1 showed different compositions between GII.4[P31] and GII.4[P16], although they were detected at approximately the same time (Table 3). The three antigenic site A sequences, namely TGSHNENN, TGSHNENH, and TGSRNEDH, of GII.4[P31] or GII.4[P16] were common in global data reported at approximately the same time [51]. Two GII.4[P16] strains in 2020 had an RTDFEVN sequence at antigenic site C, which was a minor sequence in a global report in 2020 [51]; however, this sequence was not detected in 2023. At antigenic site E, only one GII.4[P31] strain contained the SRSTP sequence, which has rarely been reported worldwide [51], but mutations in antigenic site E are less likely to be associated with the GII.4 pandemic [52]. Any of the five amino acids (residues 352, 355, 357, 368, and 378 of the VP1 amino acid sequence) were substituted in the previous pandemic GII.4 variants [9]. In the samples collected in the present study, no substitutions in these residues were found compared to those in GII.4 Sydney 2012. However, the amino acid sequence of the antigenic sites differed from that of GII.4[P31] (Table 3), and herd immunity to GII.4[P16] may have been insufficient. Investigations will continue to determine whether GII.4[P16] has been the predominant type in Osaka since 2023.

GII.2[P16], GII.4[P16], and GII.4[P31] in the present study were closely related to those collected in other regions of Japan during the same period (Figure 4C, Figure 5C, Figure 6C and Figure 7C). However, the MCC tree analysis of GII.P31 RdRp showed that 22-012 (LC798034) and 22-013 (LC798013) belonged to a different cluster, including strains collected in 2012–2015, from the other strains obtained in the present study (Figure 5C). This indicates that HuNoV strains belonging to clusters that have not been detected for a certain period may re-emerge over time and suggests that HuNoV strains will evolve differently even if they have the same type of RdRp. The GII.P31 RdRp-coding sequences of 22-012 and 22-013 completely matched each other. The identity rates of the nucleotide sequences between these two strains and the other strains obtained here ranged from 95.2% to 96.2% (including stop codons). However, the identity of the amino acid sequences ranged from 98.8% to 99.2%, with no difference in the motifs. Most of the nucleotide mutations in 22-012 and 22-013 were synonymous substitutions, suggesting that the function of RdRp was unaffected.

In previous reports, the evolutionary rates of GII.P16 RdRp were higher than those of GII.P31 (GII.Pe) [53]; however, in the present study, these rates were similar between both P-types (Figure 4C and Figure 5C). Because the Bayesian phylogenetic analysis in the present study was based solely on strains from Japan, this may not fully reflect the rates of global GII.4[P16] evolution [29]. The historical population dynamics from 2010 to 2023 remained low in Ne for GII.4, GII.P16, and GII.P31 but may be sufficient to maintain an adequate population size. Conversely, GII.2 showed a flocculating trend in Ne, which is consistent with a previous study [54].

Although several genotype/P-type combinations, such as GII.2[P16] and GII.4[P31], were commonly detected in all groups, different trends were present between children and adults. The detection rate of GII.3[P12] was higher in children (Groups A and B) than in adults or older adults (Groups C and D), whereas that of GII.17[P17] was higher in adults and older adults (Groups C and D) than in children (Groups A and B) (Figure 1). A previous study showed that the viral load in children was lower with GII.17[P17] than with GII.4[P31] [55]. This difference may provide a clue as to why GII.17[P17] is more common in adults than in children. Another study using human intestinal enteroids showed that cellular interferon responses restrict GII.3 but not GII.4 replication [56]. Thus, the biological characteristics and immune responses of HuNoV can reportedly differ depending on genotype, and these responses may be age-dependent. The present study could not determine the age dependence of each genotype of HuNoV, and larger-scale studies in the longer term are needed.

## 5. Conclusions

Although the detection of HuNoV decreased due to COVID-19 countermeasures, GII.2[P16] became prevalent among children after the second state of emergency declaration, a reminder of the high infectivity of HuNoV. Determination of HuNoV genotypes in all age groups suggested that some genotypes may be age-dependent. Phylogenetic analysis of GII.P31RdRp showed that some strains detected in 2022 belonged to different clusters than other strains obtained during the study period, suggesting that the same type of RdRp evolved differently. An analysis of the amino acid sequences of VP1 revealed differences in antigenic sites between GII.4[P16] and GII.4[P31]; this result provides important insights into the relationship between epidemic and population immunity. In the future, we will investigate the genotypes that have become prevalent and compare them to the data in the present study.

## Figures and Tables

**Figure 1 viruses-16-00654-f001:**
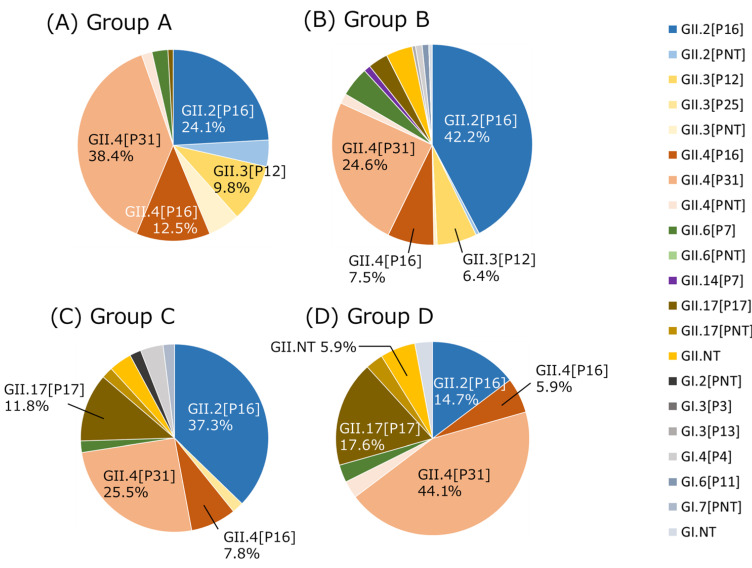
HuNoV dual typing for each group in Osaka from April 2019 to March 2023. Genotype/P-type combinations that accounted for >5% of the total are indicated by their percentage. NT stands for Non-Typable and indicates that genotype or P-type could not be determined. (**A**) Sporadic cases in children (0–14 y). (**B**) Human-to-human outbreaks in children in childcare facilities and schools. (**C**) Foodborne-suspected outbreaks. (**D**) Human-to-human outbreaks in older adults in nursing homes.

**Figure 2 viruses-16-00654-f002:**
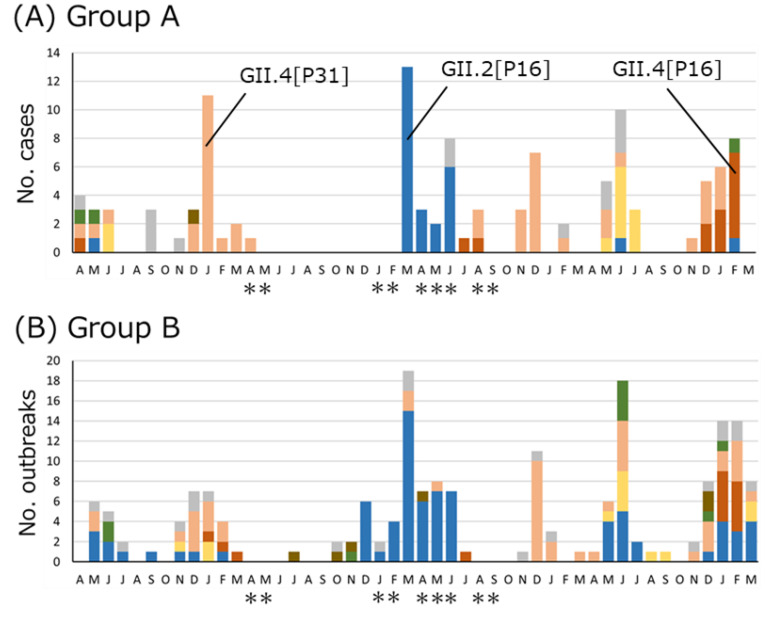
Monthly distribution of HuNoV-positive samples or outbreaks and associated genotype/P-type combinations for each group in Osaka from April 2019 to March 2023. (**A**) Sporadic cases in children (0–14 y). (**B**) Human-to-human outbreaks in children in childcare facilities and schools. (**C**) Foodborne-suspected outbreaks. (**D**) Human-to-human outbreaks in older adults in nursing homes. Months with single asterisks (*) indicate the period of implementation of the state of emergency in Osaka. These periods were: (1) 7 April–21 May 2020; (2) 14 January–28 February 2021; (3) 25 April–20 June 2021; and (4) 2 August–30 September 2021.

**Figure 3 viruses-16-00654-f003:**
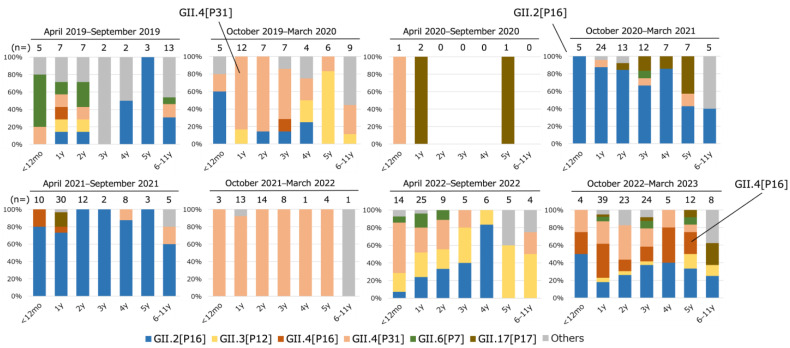
Age distribution of HuNoV genotype/P-type combinations for children in Osaka from April 2019 to March 2023. Data were collected from sporadic cases and human-to-human infection outbreaks in children aged from 0 to 11 years. Ages shown are those at the collection date. If multiple HuNoV types were found in one sample, they were all counted. In human-to-human outbreaks, if the same combination of genotype and P-type was detected in two or more samples or in >50% of the samples tested, the other positive samples were regarded as the same combination. If multiple HuNoV types were detected in one outbreak, the other samples that were not genotyped were regarded as “Others”.

**Figure 4 viruses-16-00654-f004:**
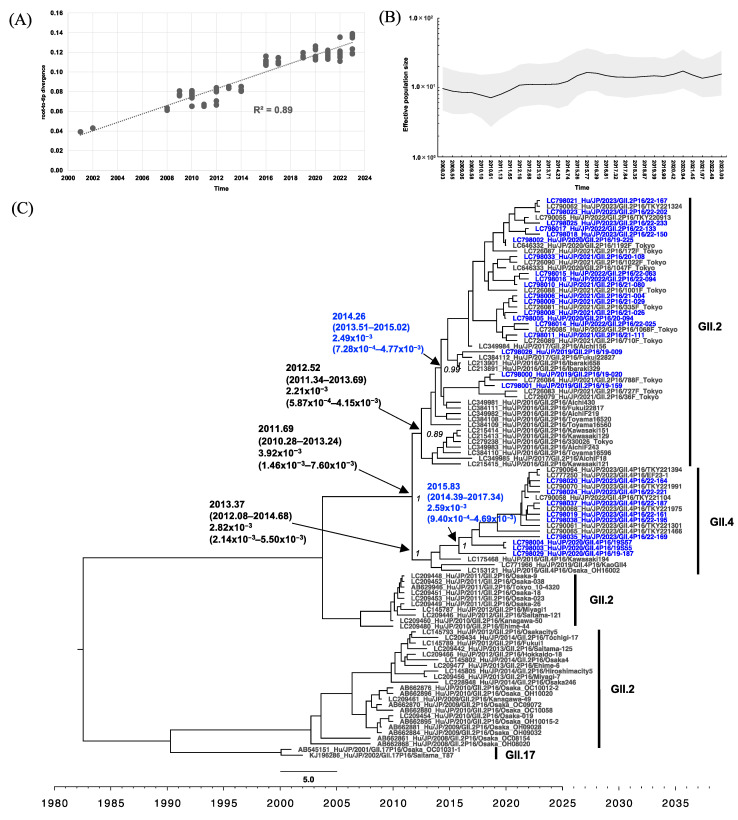
Phylogenetic analyses of HuNoV GII.P16 based on RdRp sequences from Japan. (**A**) Root-to-tip analysis. The genetic distance of each sampling sequence was plotted against the collection year; linear regression and R^2^ are shown. (**B**) Bayesian Skygrid plot illustrating the historical population dynamic. The mean of the estimated population size is represented by a black line. The 95% HPD interval is shown in the shaded area. (**C**) The time-scaled MCC tree. The tMRCA, an evolution rate with a 95% HPD interval, and posterior probability displayed at ancestral key nodes are indicated by black arrows; corresponding capsid genotypes are indicated by bracket lines, and the sequences obtained in Osaka during the present study period are indicated in blue text. Seven Osaka strains, namely LC798017, LC798021, LC798023, LC798025, LC798035, LC798037, and LC798038, were collected from the outside of the areas covered by the surveillance systems and were also included as reference strains.

**Figure 5 viruses-16-00654-f005:**
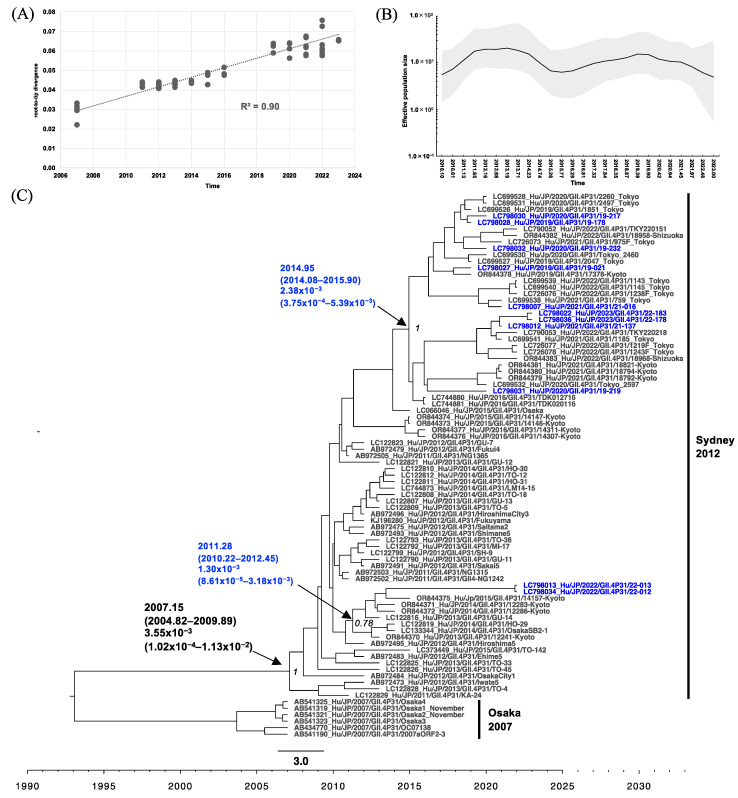
Phylogenetic analyses of HuNoV GII.P31 based on RdRp sequences from Japan. (**A**) Root-to-tip analysis. The genetic distance of each sampling sequence was plotted against the collection year; linear regression and R^2^ are shown. (**B**) Bayesian Skygrid plot illustrating the historical population dynamic. The mean of the estimated population size is represented by a black line. The 95% HPD interval is shown in the shaded area. (**C**) The time-scaled MCC tree. The tMRCA, an evolution rate with a 95% HPD interval, and posterior probability displayed at ancestral key nodes are indicated by black arrows; corresponding GII.4 variants are indicated by bracket lines, and the sequences obtained in Osaka during the present study period are indicated in blue text.

**Figure 6 viruses-16-00654-f006:**
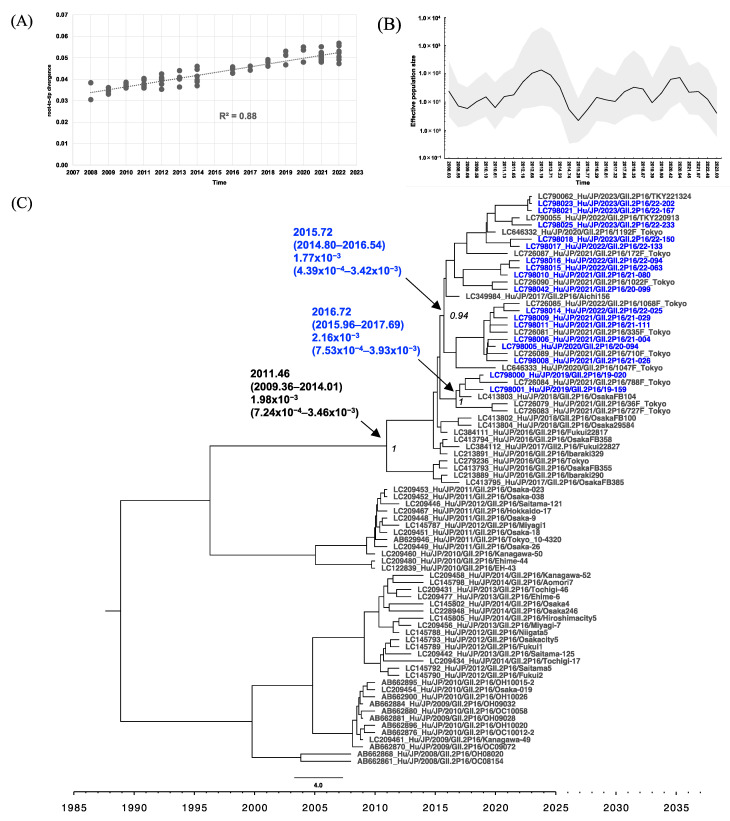
Phylogenetic analyses of HuNoV GII.2 based on VP1 sequences from Japan. (**A**) Root-to-tip analysis. The genetic distance of each sampling sequence was plotted against the collection year; linear regression and R^2^ are shown. (**B**) Bayesian Skygrid plot illustrating the historical population dynamic. The mean of the estimated population size is represented by a black line. The 95% HPD interval is shown in the shaded area. (**C**) The time-scaled MCC tree. The tMRCA, an evolution rate with a 95% HPD interval, and posterior probability displayed at ancestral key nodes are indicated by black arrows; the sequences obtained in Osaka during the present study period are indicated in blue text. Four Osaka strains, namely LC798017, LC798021, LC798023, and LC798025, were collected from the outside of the areas covered by the surveillance systems and were also included as reference strains.

**Figure 7 viruses-16-00654-f007:**
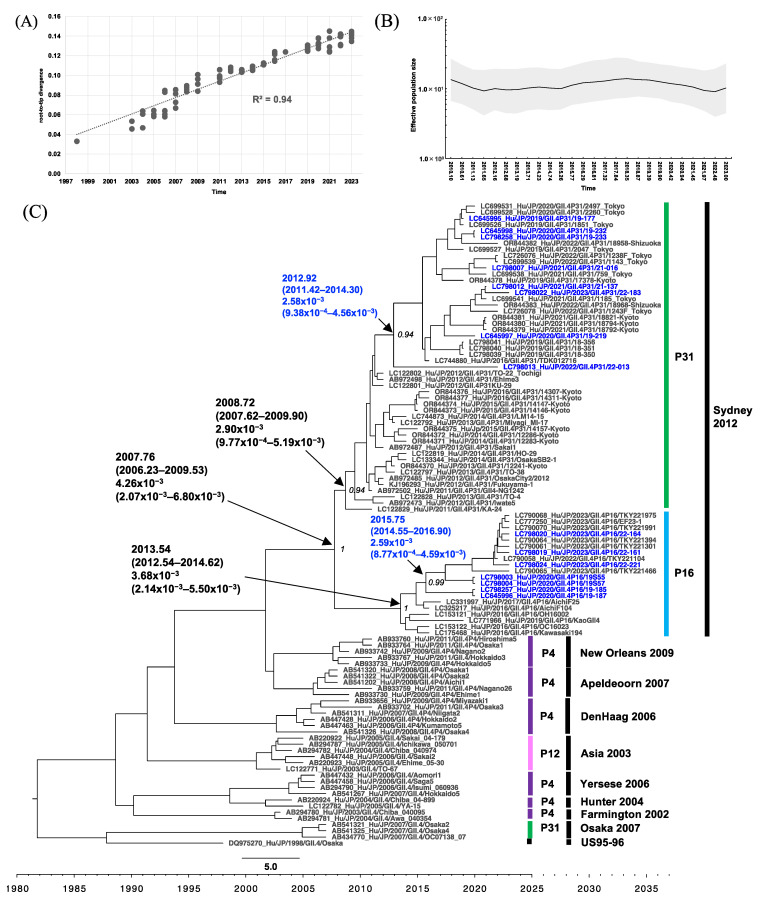
Phylogenetic analyses of HuNoV GII.4 based on VP1 sequences from Japan. (**A**) Root-to-tip analysis. The genetic distance of each sampling sequence was plotted against the collection year; linear regression and R^2^ are shown. (**B**) Bayesian Skygrid plot illustrating the historical population dynamic. The mean of the estimated population size is represented by a black line. The 95% HPD interval is shown in the shaded area. (**C**) The time-scaled MCC tree. The tMRCA, an evolution rate with a 95% HPD interval, and posterior probability displayed at ancestral key nodes are indicated by black arrows. Corresponding P-types are indicated by bracket lines in different colors, and corresponding GII.4 variants are indicated by bracket lines in black; the sequences obtained in Osaka during the present study period are indicated in blue text. Three Osaka strains, namely LC798039–LC798041, were collected in Osaka before the present study period and were also included as reference strains.

**Table 1 viruses-16-00654-t001:** Grouping based on mode of infection for samples collected from April 2019 to March 2023.

Group	Infection Route	Specifications	No. of Outbreaks	No. of HuNoV Outbreaks (%)	No. of Samples	No. of Positives for HuNoV (%)
A	Sporadic	0–14 y	-	-	326	112 (34.4)
B	Human-to-human	Childcare facilities and schools	258	182 (70.5)	801	441 (55.1)
C	Suspected foodborne	-	139	50 (36.0)	690	185 (26.8)
D	Human-to-human	Nursing homes	34	32 (94.1)	105	89 (84.8)

**Table 2 viruses-16-00654-t002:** Analysis of GII.P16 RdRp amino acid substitutions of HuNoV strains detected in the present study compared to the reference strains.

	Type	Name of Strain	Accession Number	Collection Year	Amino Acid Position																						
					14	18	54	68	81	85	111	121	125	132	175	178	208	257	274	312	386	405	427	457	464	479	502
References	GII.2 [P16]	Santa Rosa 1764	KY865306	2016	L	G	K	K	S	A	A	H	V	N	I	K	A	A	I	A	N	K	N	K	T	E	N
	GII.4 [P16]	SH21-668	OQ940080	2021	L	G	R	K	S	A	A	H	A	N	I	K	A	A	I	T	N	K	T	R	T	E	N
Samples in the Present Study	GII.2 [P16]	19-009	LC798026	2019	L	G	K	K	S	A	A	H	V	N	I	K	A	A	T	A	N	K	N	K	T	E	N
		19-020	LC798000	2019	L	G	K	K	G	A	A	H	V	N	V	K	T	A	I	A	N	K	N	K	T	E	N
		19-159	LC798001	2019	L	G	K	K	G	A	A	H	V	N	V	K	A	A	I	A	D	K	N	K	T	E	N
		19-225	LC798002	2020	L	G	K	K	S	A	A	H	V	N	I	K	A	A	T	A	N	K	N	K	T	E	N
		20-094	LC798005	2020	L	G	K	K	S	A	A	H	V	N	I	K	A	A	I	A	N	K	N	K	T	E	N
		20-108	LC798033	2021	L	G	K	K	S	A	A	H	V	N	I	K	A	A	I	A	N	K	N	K	T	E	N
		21-004	LC798006	2021	I	G	K	K	S	A	A	H	V	N	I	K	A	A	I	A	N	K	N	K	T	E	N
		21-026	LC798008	2021	L	D	K	R	S	A	A	Y	V	N	I	K	A	A	I	A	N	K	N	K	N	E	N
		21-029	LC798009	2021	L	G	K	K	S	A	A	H	V	N	I	K	A	A	I	A	N	K	N	K	T	E	N
		21-080	LC798010	2021	L	G	K	K	S	A	V	H	V	N	I	K	A	A	I	A	N	K	N	K	T	E	N
		21-111	LC798011	2021	L	G	K	K	S	A	A	H	V	N	I	K	A	A	I	A	N	K	N	K	T	E	N
		22-025	LC798014	2022	L	G	K	K	S	A	A	H	V	N	I	K	A	T	I	A	N	K	N	K	T	E	N
		22-063	LC798015	2022	L	G	K	K	S	A	V	H	V	N	I	K	A	A	I	A	N	K	N	K	T	E	N
		22-094	LC798016	2022	L	G	K	K	S	A	V	H	V	N	I	K	A	A	I	A	N	K	N	K	T	E	N
		22-150	LC798018	2023	L	G	K	K	S	S	A	Y	V	N	I	K	A	A	T	A	N	R	N	K	T	E	N
	GII.4 [P16]	19-187	LC798029	2020	L	S	K	K	S	A	A	H	V	N	I	R	A	A	I	A	N	K	S	R	T	E	S
		19S55	LC798003	2020	L	G	R	K	S	A	A	H	A	N	I	K	A	A	I	T	N	K	N	R	T	E	N
		19S57	LC798004	2020	L	G	R	K	S	A	A	H	A	N	I	K	A	A	I	T	N	K	N	R	T	E	N
		22-161	LC798019	2023	L	G	R	K	S	A	A	H	V	S	I	K	T	A	I	A	D	K	N	R	T	G	N
		22-164	LC798020	2023	L	G	R	K	S	A	A	H	V	S	I	K	A	A	I	A	D	K	N	R	T	E	N
		22-221	LC798024	2023	L	G	R	K	S	A	A	H	V	S	I	K	A	A	I	A	D	K	N	R	T	E	N

Note: Amino acid residues that differ from those of KY865306 were highlighted in blue.

**Table 3 viruses-16-00654-t003:** Analysis of the VP1 antigenic sites of GII.4 noroviruses detected in the present study.

	Type	Name of Variant or Strain	Accession Number	Collection Year	Amino Acid Position																														
					Antigenic Site A								Antigenic Site C							Antigenic Site D					Antigenic Site E					Antigenic Site G					
					294	295	296	297	298	368	372	373	339	340	341	375	376	377	378	393	394	395	396	397	407	411	412	413	414	352	355	356	357	359	364
References	GII.4[P31]	Sydney 2012_NSW0514	JX459908	2012	T	G	S	R	N	E	D	R	R	T	D	F	E	A	N	G	T	T	H	R	S	R	N	T	H	Y	S	A	D	A	R
	GII.4[P16]	Sydney 2012_OH16002	LC153121	2016	T	G	S	R	N	E	D	H	R	T	D	F	E	A	N	S	T	T	H	R	S	R	N	T	H	Y	S	A	D	A	R
	GII.4[P16]	SH21-668	OQ940080	2021	T	G	S	H	N	E	N	H	R	T	D	F	E	A	N	S	T	T	H	R	S	R	N	T	H	Y	S	A	D	A	R
Samples in the Present Study	GII.4[P31]	19-177	LC645995	2019	T	G	S	H	N	E	N	N	R	T	D	F	E	A	N	G	T	T	H	R	S	R	N	T	P	Y	S	A	D	A	R
		19-219	LC645997	2020	T	G	S	H	N	E	N	N	R	T	D	F	E	A	N	G	T	T	H	R	S	R	N	T	P	Y	S	A	D	A	R
		19-232 *	LC645998	2020	T	G	S	H	N	E	N	N	R	T	D	F	E	A	N	G	T	T	H	R	S	R	N	T	P	Y	S	A	D	A	R
		19-233 *	LC798258	2020	T	G	S	H	N	E	N	N	R	T	D	F	E	A	N	G	T	T	H	R	S	R	N	T	P	Y	S	A	D	A	R
		21-016	LC798007	2021	T	G	S	H	N	E	N	N	R	T	D	F	E	A	N	G	T	T	H	R	S	R	N	T	P	Y	S	A	D	A	R
		21-137	LC798012	2021	T	G	S	H	N	E	N	N	R	T	D	F	E	A	N	G	T	T	H	R	S	R	N	T	P	Y	S	A	D	A	R
		22-013	LC798013	2022	T	G	S	H	N	E	N	H	R	T	D	F	E	A	N	G	T	T	H	R	S	R	S	T	P	Y	S	A	D	A	R
		22-183	LC798022	2023	T	G	S	H	N	E	N	N	R	T	D	F	E	A	N	G	T	T	H	R	S	R	N	T	P	Y	S	A	D	A	R
	GII.4[P16]	19-185 *	LC798257	2020	T	G	S	R	N	E	D	H	R	T	D	F	E	V	N	S	T	T	H	R	S	R	N	T	H	Y	S	A	D	A	R
		19-187 *	LC645996	2020	T	G	S	R	N	E	D	H	R	T	D	F	E	V	N	S	T	T	H	R	S	R	N	T	H	Y	S	A	D	A	R
		19S55	LC798003	2020	T	G	S	H	N	E	N	H	R	T	D	F	E	A	N	S	T	T	H	R	S	R	N	T	H	Y	S	A	D	A	R
		19S57	LC798004	2020	T	G	S	H	N	E	N	H	R	T	D	F	E	A	N	S	T	T	H	R	S	R	N	T	H	Y	S	A	D	A	R
		22-161	LC798019	2023	T	G	S	H	N	E	N	H	R	T	D	F	E	A	N	S	T	T	H	R	S	R	N	T	H	Y	S	A	D	A	R
		22-164	LC798020	2023	T	G	S	H	N	E	N	H	R	T	D	F	E	A	N	S	T	T	H	R	S	R	N	T	H	Y	S	A	D	A	R
		22-221	LC798024	2023	T	G	S	H	N	E	N	H	R	T	D	F	E	A	N	S	T	T	H	R	S	R	N	T	H	Y	S	A	D	A	R

Note: All antigenic sites were reported as conformational epitopes [12]. *: from the same outbreak.

## Data Availability

The data presented in the present study are available upon request from the corresponding author. The data are not publicly available due to privacy and ethical restrictions. Nucleotide sequences are available in the GenBank database under the deposited numbers cited within the manuscript.

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
