# Peer review of "Epidemiological Features of Human Norovirus Genotypes before and after COVID-19 Countermeasures in Osaka, Japan"

_viruses, 2024, doi:10.3390/v16040654_

Round 1
Reviewer 1 Report
Comments and Suggestions for Authors
In this manuscript, authors tried to detect and compare the genomes of human norovirus before and after COVID-19 countermeasures in Osaka. It is valuable to evaluate the evolution of human norovirus. The results demonstrated that new genotypes were dominated. However, some comments were as follows.
1, It seemed that authors were not familiar with the difference between norovirus and human norovirus. Line 36. To our knowledge, the genome of murine norovirus comprises FOUR ORFs. Line 39. Noroviruses are classified into 10 genogroups (GI–GX). Do all norovirus strains cause acute gastroenteritis? So, it would be corrected in the whole draft.
2, The introduction was wordy.
3, Line 88. The detail information of samples was missed. Epidemiological features of norovirus should involve many things, such as patient gender, age, etc.
4, Line 99. In this section, the new design primers and probes were reported. Why did authors not use them to detect norovirus? Did the primers and probes design by authors? No refs?
Comments on the Quality of English Language
The draft should be revised by a native speaker.
Reviewer 2 Report
Comments and Suggestions for Authors
Dear Authors
Thank you for the well drafted manuscript. It was pleasant to read. Please see minors corrections and recommendations below.
1. Pg 3 of 20, Line 126:
Is GII.4P31_5132R the correct primer for amplifying the RdRp region of GII.4[P16] strains?
2. All figures in the main text, especially the phylogenetic trees are too small and not easy to examine. While the trees in the supplementary section are slightly larger, their presentation can also be improved
Reviewer 3 Report
Comments and Suggestions for Authors
REVIEW REPORT
The authors report the epidemiological features of Norovirus genotypes before and after the pandemic COVID-19. They used the phylogenetic and recombination analyses to compare the sequences. They observed changes in the circulating genotypes.
The paper is worthy to be considered for publication provide the authors addressed the suggested recommendations and corrections.
1. Title
This could be improved by adding the word genotypes: “Epidemiological features of Norovirus genotypes before and after COVID-19 countermeasures in Osaka, Japan”.
2. Abstract and Introduction
The study rationale is not clearly stated in this report (in the abstract and introduction).
3. Materials and Methods
3.1. -Where are the exact times (Months) of sample collections? This is important as the circulation of enteric viruses vary according to the seasons.
-Describe the aspect of stools collected. Were they all diarrhea cases?
3.2. Detection of Norovirus
Line 101: …saline then thoroughly mixed via vortexing and clarified…
3.3. -Line 116 and 120: Give the exact number of positive samples.
-List the size of PCR products in the sequencing and genotyping report.
-How many samples were successfully sequenced?
3.4. Statistical analysis must be added as universally recommended in a
scientific publication.
4. Results
-Line 177: Remove the word “results”.
-Table 1: The period of samples collection should be mentioned in the title.
5. Discussion and conclusion
-This is a paper and not a dissertation. The two first paragraphs are unnecessary. It could be only a summary of results there.
-Line 391: …in 2021 compared to those…
-Line 415: Ne of GII P16 … What does it mean?
I suggest the study authors to give the impact of this research study and findings. Why are they important to be considered for continuous investigation? This is missing even in the abstract of the report.
Supplementary figures
Figures S1, S2, S3 and S4:
In the captions, data on the size of partial fragments of RdRp and VP1 analyzed are missing. The number of referenced sequences is missing as well.
Comments on the Quality of English LanguageThe English is acceptable for publication.
Round 2
Reviewer 1 Report
Comments and Suggestions for Authors
I believed that authors focused on the detection of human norovirus before and after COVID-19. So, it is necessary to modify the title and whole manuscript.
BTW, GenBank is the correct format.
Comments on the Quality of English LanguageI believed that authors focused on the detection of human norovirus before and after COVID-19. So, it is necessary to modify the title and whole manuscript.
BTW, GenBank is the correct format.
